# Potential of Tyrosine Kinase Receptor TIE-1 as Novel Therapeutic Target in High-PI3K-Expressing Ovarian Cancer

**DOI:** 10.3390/cancers12061705

**Published:** 2020-06-26

**Authors:** Xuewei Zhang, Masumi Ishibashi, Kazuyuki Kitatani, Shogo Shigeta, Hideki Tokunaga, Masafumi Toyoshima, Muneaki Shimada, Nobuo Yaegashi

**Affiliations:** 1Department of Obstetrics and Gynecology, Tohoku University Graduate School of Medicine, Tohoku University, Sendai 980-8577, Japan; zhangxuejian9521@yahoo.co.jp (X.Z.); s.shigeta@med.tohoku.ac.jp (S.S.); tokunagahideki@med.tohoku.ac.jp (H.T.); mshimada1221@yahoo.co.jp (M.S.); yaegashi@med.tohoku.ac.jp (N.Y.); 2Cancer Science Institute of Singapore, National University of Singapore, Singapore 119077, Singapore; 3Laboratory of Immunopharmacology, Faculty of Pharmaceutical Sciences, Setsunan University, Hirakata, Osaka 572-8508, Japan; kazuyuki.kitatani@pharm.setsunan.ac.jp; 4Department of Obstetrics and Gynecology, Japanese Red Cross Ishinomaki Hospital, Ishinomaki 986-8522, Japan; m-toyo@med.tohoku.ac.jp

**Keywords:** TIE-1, PI3K, ovarian cancer, molecular targeted therapy

## Abstract

Tyrosine kinase receptor TIE-1 plays a critical role in angiogenesis and blood-vessel stability. In recent years, increased TIE-1 expression has been observed in many types of cancers; however, the biological significance and underlying mechanisms remain unknown. Thus, in the present study, we investigated the tumor biological functions of TIE-1 in ovarian cancer. The treatment of SKOV3 ovarian-cancer cells with siRNA against TIE-1 decreased the expression of key molecules in the PI3K/Akt signaling pathway, such as p110α and phospho-Akt, suggesting that TIE-1 is related to the PI3K/Akt pathway. Furthermore, the knockdown of TIE-1 significantly decreased cell proliferation in high-PI3K-expressing cell lines (SKOV3, CAOV3) but not low-PI3K-expressing cell lines (TOV112D, A2780). These results suggested that inhibition of TIE-1 decreases cell growth in high-PI3K-expressing cells. Moreover, in low-PI3K-expressing TOV112D ovarian-cancer cells, TIE-1 overexpression induced PI3K upregulation and promoted a PI3K-mediated cell proliferative phenotype. Mechanistically, TIE-1 participates in cell growth and proliferation by regulating the PI3K/Akt signaling pathway. Taken together, our findings strongly implicate TIE-1 as a novel therapeutic target in high-PI3K-expressing ovarian-cancer cells.

## 1. Introduction

Ovarian cancer is a highly lethal global gynecologic malignancy [1,2]. Most patients are diagnosed at an advanced stage, and the five-year survival rate is in the range of 30–40% [3]. Although ovarian-cancer cells are sensitive to platinum-based chemotherapy, recurrence and the development of drug resistance make ovarian cancer difficult to treat [4]. Thus, the establishment of novel therapeutic strategies for refractory ovarian cancer is an urgent need.

Tyrosine kinase with immunoglobulin-like and EGF-like domains (TIE)-1 is a cell-membrane protein expressed in endothelial cells [5,6]. Although TIE-1 is considered an orphan receptor thus far [7,8], recently LECT2, a functional ligand of TIE-1, has been reported [9]. Xu and colleagues showed that LECT2/TIE-1 signaling pathway promotes liver fibrogenesis; however, the significance of LECT2/TIE-1 in cancer is still not clear. The known functions of TIE-1 are in angiogenesis and blood-vessel stability [10,11]. Recently, it was reported that TIE-1 upregulation predicts shorter survival in patients with early chronic-phase chronic myeloid leukemia [12,13]. Moreover, TIE-1 overexpression was observed in many types of cancer, including gastric- [14,15], colon- [16], and breast-cancer cells [17], but the significance of TIE-1 overexpression in cancer remains unknown. We previously reported that TIE-1 promotes DNA damage repair, thereby rendering ovarian-cancer cells resistant to cisplatin [18]. Therefore, TIE-1 might be a potential target for anticancer treatment.

Phosphoinositide 3-kinases (PI3Ks) are members of the lipid-kinase family originally discovered in the 1980s [19,20]. On the basis of their structure and substrate specificity, PI3Ks are categorized into three distinct classes: I, II, and III [21,22]. Class I is divided into Class IA and IB on the basis of differences in activating receptors [23,24]. In Class IA, PI3Ks are heterodimers consisting of a p85 regulatory subunit and a p110 catalytic subunit that has three isomers (α, β, and γ) encoded by three distinct genes, *PIK3CA*, *PIK3CB*, and *PIK3CD*, respectively. Of these, *PIK3CA* is the most frequently mutated in human cancer [25,26].

The PI3K/Akt pathway is one of the most essential intracellular pathways that mediates cellular metabolism, survival, differentiation, and cell growth [27,28]. Recent evidence suggests that different kinds of receptor tyrosine kinases (RTKs) are associated with the activation of the PI3K/Akt signaling pathway [29,30] that promotes cell proliferation, differentiation, migration, and survival. However, the involvement of TIE-1, one of the RTKs, in PI3K/Akt pathway remains unknown and requires further research.

Therefore, in the present study, we explore the biological significance of TIE-1 in the PI3K/Akt signaling pathway and demonstrate a novel molecular mechanism of action for TIE-1 in ovarian cancer, identifying TIE-1 as a target for the treatment of high-PI3K-expressing ovarian cancer.

## 2. Results

### 2.1. TIE-1 Might Signal through PI3K/Akt Pathway

To determine the biological functions of tyrosine kinase receptor TIE-1, we introduced TIE-1 knockdown using siRNA and measured the expression of key proteins in the PI3K/Akt signaling pathway. Immunoblotting analysis showed that TIE-1 knockdown significantly suppressed the protein expression of PI3K p110α and phospho-Akt, with no change in total Akt (Appendix A), in SKOV3 ovarian cancer cells (Figure 1A–C). Immunoblotting analysis showed that overexpression of TIE-1 with a V5-tagged TIE-1 vector significantly enhanced the protein expression of PI3K p110α and phospho-Akt, with no change in total Akt (Appendix A), in SKOV3 cells (Figure 1D–F). These results suggested that TIE-1 might signal through the PI3K/Akt signaling pathway.

### 2.2. TIE-1 Inhibition Decreases Cell Growth in High-PI3K-Expressing Cell Line.

TIE-1 and PI3K expression levels in 11 ovarian-cancer cell lines were confirmed. TIE-1 and PI3K p110α expression varied in the tested ovarian-cancer cell lines (Figure 2A,C,D). There was positive-correlation tendency between the protein-expression levels of TIE-1 and PI3K p110α (Figure 2B). However, there was no significant difference (*p* > 0.05). We next investigated the effect of TIE-1 knockdown on cell growth using multiple ovarian-cancer cell lines. On the basis of the endogenous expression of PI3K in cells and/or whether TIE-1 knockdown induced cell-growth inhibition, we divided 11 ovarian-cancer cell lines into two groups: low-PI3K-expressing and high-PI3K-expressing cell lines (Figure 2C, Appendix A). Low-PI3K-expressing cell lines TOV112D, or A2780 and high-PI3K-expressing cell lines SKOV3 or CAOV3, were used. Treatment with TIE-1 siRNA for 120 h significantly reduced cell number by approximately 45% in high-PI3K-expressing SKOV3 cells (Figure 2H), and by approximately 33% in CAOV3 cells (Figure 2I) relative to control siRNA group. Interestingly, the knockdown of TIE-1 significantly decreased cell proliferation in high-PI3K-expressing cell lines SKOV3 and CAOV3 (Figure 2E,H,I), but not in low-PI3K-expressing cell lines TOV112D and A2780 (Figure 2E–G). These results suggested that inhibition of TIE-1 selectively suppresses PI3K/Akt-mediated cell growth in high-PI3K-expressing cells.

### 2.3. TIE-1 Inhibition Induces Apoptosis in High-PI3K-Expressing Cell Lines

Given that PI3K inhibitor GDC-0941 is a potent inducer of apoptosis (Appendix A), we determined whether TIE-1 inhibition-induced PI3K downregulation triggers apoptosis. To elucidate the underlying molecular mechanism of TIE-1 inhibition-induced cell death, we performed an annexin V/7-annexin-apoptosis-detection (AAD) assay. TIE-1 inhibition had no effect on the percentage of apoptotic cells in low-PI3K-expressing TOV112D (Figure 3A,B) or A2780 (Figure 3C,D) cells. In contrast, TIE-1 inhibition increased the percentage of apoptotic high-PI3K-expressing SKOV3 (Figure 3E,F) and CAOV3 (Figure 3G,H) cells.

Furthermore, cleaved poly ADP-ribose polymerase (PARP) and cleaved caspase-3 expression levels were significantly increased in TIE-1-deficient high-PI3K-expressing SKOV3 cells (Figure 3I). In addition, procaspase-3 expression was decreased in TIE-1 siRNA-treated SKOV3 cells (Figure 3I). Similar results were observed in CAOV3 ovarian-cancer cells (Appendix A). These findings suggest that inhibition of TIE-1 induces apoptosis in high- but not low-PI3K-expressing cells. Therefore, TIE-1 may function as an antiapoptotic regulator in high-PI3K-expressing ovarian-cancer cells.

### 2.4. Increased TIE-1 Expression Induces TIE-1/PI3K-Mediated Cell Growth in Low-PI3K-Expressing Ovarian-Cancer Cells

To determine the mechanism by which TIE-1 selectively exhibits an antitumor effect in high-PI3K-expressing cells, we investigated whether high-PI3K-expressing cells showed PI3K-mediated cell growth. We transfected low-PI3K-expressing TOV112D cells with a PIK3CA–Flag vector, and measured cell growth.

As shown in Figure 4A, transfection with PIK3CA–Flag vectors was confirmed to increase PI3K protein expression. Importantly, PIK3CA–Flag-overexpressing significantly promoted cell growth compared to empty vectors (Figure 4B). We treated PIK3CA–Flag stably overexpressing TOV112D cells with a PI3K inhibitor (GDC-0941), and determined its IC50 value. IC_50_ in control cells or PI3K p110α-overexpressed cell number was 5.23 or 2.41 µM (Figure 4C), respectively. These findings suggested that PI3K upregulation promotes PI3K-mediated cell growth in low-PI3K-expressing cells.

Given our data demonstrating that PI3K upregulation induces PI3K-mediated cell growth, V5-tagged TIE-1 vector-induced PI3K upregulation may also promote PI3K-mediated cell growth. To confirm this hypothesis, we established a V5-tagged TIE-1 stably overexpressing TOV112D cell line and measured TIE-1 and PI3K expression, and cell growth. In V5-tagged TIE-1 stably overexpressing TOV112D cells, high TIE-1 levels promoted PI3K expression and cell proliferation compared with empty V5 vector cells (Figure 4D,E). The IC_50_ of the PI3K inhibitor (GDC-0941) in empty V5 vector cells was 7.93 µM (Figure 4F). In contrast, the IC_50_ in TIE-1-overexpressed cells was 1.75 µM (Figure 4F). These results suggested that TIE-1 overexpressed cells were more sensitive to the PI3K inhibitor compared with empty-vector cells.

Then, we treated empty V5 vector cells or V5-tagged TIE-1 stably overexpressing TOV112D cells with PI3K p110α siRNA and measured cell viability. The knockdown of PI3K in both V5-tagged TIE-1 stably overexpressing TOV112D cells and empty V5 vector cells significantly inhibited cell growth (Figure 4G,H). Moreover, in V5-tagged TIE-1 stably overexpressing TOV112D cells, inhibition of PI3K significantly decreased cell proliferation (Figure 4H). Consistent with Figure 4H, clonogenic-assay results suggested that PI3K inhibition significantly induced cell death in TIE-1-overexpressed cells compared with empty-vector cells (Figure 4I). These findings suggested that PI3K inhibition abolished high-TIE-1-expression-induced cell growth. Furthermore, TIE-1 overexpression increased PI3K expression and promoted a PI3K-mediated cell-proliferative phenotype (Figure 5).

## 3. Discussion

To the best of our knowledge, we demonstrated for the first time that tyrosine kinase receptor TIE-1 participates in cell growth and proliferation by targeting the PI3K/Akt signaling pathway in high-PI3K-expressing ovarian cancer cells. As shown in Table 1, various histological subtypes of ovarian-cancer cells were used in this study (Cancer Cell Line Encyclopedia: https://portals.broadinstitute.org/ccle). The effect of TIE-1 inhibition depended on the high level of PI3K expression in ovarian-cancer cells, not histological types. These findings provide evidence that TIE-1 may be a novel therapeutic target in high-PI3K-expressing ovarian cancer.

Endothelial cell-surface protein TIE-1 was considered as a nonfunctional orphan receptor except for the maintenance of vascular integrity [31,32]. Recently, Kontos CD et al. showed that TIE-1 activates the PI3K/Akt signaling pathway to inhibit UV-irradiation-induced apoptosis [33]. Our results suggested that TIE-1 regulates the PI3K/Akt pathway by increasing/decreasing PI3K expression in ovarian cancer. However, there is no significant difference between the protein-expression levels of TIE-1 and PI3K p110α. Considering that PIK3CA mutations were reported not only in SKOV3 but also in A2780 ovarian-cancer cells [34], it was inferred that TIE-1 and PI3K p110α expression levels might not always correlate because PIK3CA is independently mutated/amplified in some ovarian-cancer cells. Although the PI3K/Akt signaling pathway is proposed to be a downstream target of tyrosine kinase receptor TIE-1, whether PI3K is directly targeted by TIE-1 remains to be investigated. Our ongoing studies aim to identify the transcription factors responsible for the TIE-1-mediated regulation of PI3K.

The PI3K/Akt/mTOR pathway is altered in approximately 70% of ovarian-cancer cases, leading to enhanced cellular growth, proliferation, and survival, through an intricate series of hyperactive signaling cascades [35,36]. This pathway activation is associated with higher invasive and migratory capacities in human ovarian cancer; thus, the PI3K/Akt/mTOR pathway is a potential predictor of invasiveness in ovarian-tumor cells [37]. Therefore, the inhibition of this pathway represents a potential therapeutic strategy for cancer treatment [38,39]. However, due to the nonspecific growth inhibition of normal cells and other adverse effects, the use of oral selective PI3K inhibitors remains challenging [40,41]. Our findings demonstrated that TIE-1 inhibition selectively exhibited an antitumor effect by decreasing PI3K expression in high-PI3K-expressing ovarian-cancer cells. Furthermore, TIE-1 inhibition-induced PI3K downregulation did not affect low-PI3K-expressing ovarian-cancer or normal cells. These data suggested that TIE-1 inhibition-based therapy may provide an alternative approach to PI3K/Akt pathway inhibitors for the treatment of cancers with high PI3K expression.

In summary, TIE-1 functions as a potential carcinogenic molecule by activating the PI3K/Akt signaling pathway in high-PI3K-expressing ovarian cancer. Therefore, the future development of specific TIE-1 inhibitors could be an effective strategy to improve the prognosis of ovarian-cancer patients. Furthermore, PI3K could serve as a possible biomarker to predict the therapeutic efficiency of TIE-1-inhibition-based therapy in the future.

## 4. Materials and Methods

### 4.1. Antibodies and Reagents

Peroxidase AffiniPure Donkey Anti-Rabbit IgG (#711-035-152) and Peroxidase AffiniPure Donkey Anti-Mouse IgG (#715-035-150) were from Jackson ImmunoResearch (West Grove, PA, USA). Antibodies specific to TIE-1 (ab111547) and PTEN (ab32199) were obtained from Abcam (Cambridge, MA, USA). Other antibodies for phospho-Akt (S473, #9271) and PI3K p110α (#4249) were from Cell Signaling Technology (Danvers, MA, USA). Antibodies specific to V5 (#R960-25) were obtained from Thermo Fisher Scientific (Waltham, MA, USA). The β-actin antibody (A5441) was obtained from Sigma (St Louis, MO, USA). RNAiMax; Lipofectamine 2000; and siRNAs for control (4390846), TIE-1 (s14140 and s14141), and PI3K (s10521) were from Life Technologies (Carlsbad, CA, USA). The CellTiter-Glo luminescent cell-viability-assay kit was obtained from Promega (Fitchburg, WI, USA). The annexin V/7-AAD kit was from BD Biosciences (San Jose, CA, USA).

### 4.2. Cell Culture

Ovarian-cancer cells were cultured in Dulbecco’s modified Eagle’s medium, supplemented with 10% fetal bovine serum at 37 °C in a humidified incubator containing 5% CO_2_. Human ovarian cancer cell lines (TOV112D, CAOV3, SKOV3, A2780, A2780CP, PE01, PE04, and ES2) were obtained from the American Type Culture Collection. JHOC5, JHOC7, and JHOC8 human ovarian cancer cell lines were kindly provided by Dr. Katsutoshi Oda (University of Tokyo, Tokyo, Japan).

### 4.3. Preparation of TIE-1–V5 Vector

The preparation of TIE-1–V5-vector was previously described by Ishibashi et al. [18].

### 4.4. siRNA transfections

Cells were transfected with 5 nM of siRNAs using Lipofectamine RNAiMAX transfection reagent (Life Technologies) according to the manufacturer’s instructions.

### 4.5. Vector transfections

Cells were transfected with a TIE-1–V5 or empty vector (2 μg/dish) using Lipofectamine 2000 transfection reagent (Life Technologies) according to the manufacturer’s instructions.

### 4.6. Immunoblotting

Cells (2 × 10^5^ cells/well) were seeded in 60 mm dishes and treated with siRNAs or vectors for 72 h. Cells were harvested, washed three times with ice-cold phosphate-buffered saline (PBS), and lysed in RIPA lysis buffer. The protein content of the samples was measured using the BCA protein assay reagent. Equal amounts of proteins were subjected to SDS-PAGE (4–20% gradient gels). Proteins were electrophoretically transferred to nitrocellulose membranes and blocked with PBS/0.1% Tween 20 (PBS-T) containing 5% nonfat dried milk for 30 min at room temperature. Membranes were incubated overnight at 4 °C with the following primary antibodies: TIE-1 (1:500), PI3K p110α (1:1000), phospho-Akt (S473) (1: 1000), PTEN (1:500), and V5 (1:5000). β-actin (1:200,000) was used as a loading control. After washing three times with PBS-T, membranes were incubated with a secondary antibody conjugated with horseradish peroxidase in PBS-T containing 5% nonfat dried milk for 1 h at 4 °C. Proteins were detected using SuperSignal West Dura Extended Duration Substrate and ChemiDoc^TM^ MP (Bio-Rad, Hercules, CA, USA).

### 4.7. Cell-Viability Assay

Cells (1 × 10^3^ cells/well) were plated into a 96-well plate, followed by growth for 24 h in a humidified 37 °C cell-culture incubator. Next, cells were treated with 0.01, 0.03, 0.1, 0.3, 1, 3, 10, or 30 µM of PI3K inhibitor GDC-0941 (Pictilisib). Cell viability was determined using a CellTiter-Glo luminescent cell-viability assay (Promega) according to the manufacturer’s protocol.

### 4.8. Quantitative Real-Time PCR

Total RNA was extracted after the homogenization of cell samples using the QIAshredder and RNeasy mini kits (Qiagen, Hilden, Germany). Extracted RNAs were reverse-transcribed with SuperScript III (Thermo Fisher Scientific) for cDNA synthesis. GAPDH was used as the input reference. Real-time PCR for determining TIE-1 and PI3K mRNA expression was performed using the StepOne Plus Real-Time PCR System with Taqman Universal Master Mix II and a TaqMan probe specific for each gene (Thermo Fisher Scientific).

### 4.9. Apoptosis Assay (Annexin V/7-AAD)

To determine the proportion of apoptotic cells, the annexin V apoptosis-detection kit was used according to the manufacturer’s protocol. Each cell line was treated with control siRNA, TIE-1 siRNA, DMSO (negative control), or PI3K inhibitor GDC-0941. Cells were trypsinized and collected by centrifugation. After removing the supernatant, cells were washed with PBS and resuspended in annexin V binding buffer. Cells were labeled with 2.5 μL of annexin V-FITC and 1 µL of 7-AAD in 50 μL annexin V binding buffer, and incubated in the dark at room temperature for 30 min. Apoptotic cells were then detected by flow cytometer. The percentage of cells undergoing apoptosis was quantified. Cells are considered necrotic if they allow the penetration of 7-AAD without annexin V staining.

### 4.10. Clonogenic Assay

Cells were trypsinized, and 2 × 10^4^ cells/dish were seeded in 6 cm culture dishes. The following day, cells were treated with control siRNA or PI3K p110α siRNA for 9 days. Colonies were fixed in 4% formaldehyde for 30 min at room temperature. Fixed cells were stained with 0.5% crystal violet in ethanol for 10 min, and all images were captured using ApeosPort-VI C4471 (Fuji Xerox, Tokyo, Japan).

### 4.11. Trypan-Blue Exclusion Assay

Cells were plated into six-well plates (5 × 10^4^ cells/well) and preincubated with growth medium overnight. The following day, cells were treated with control siRNA or TIE-1 siRNA for up to 120 h. Trypan-blue-negative living cells were counted.

### 4.12. Statistical Analysis

Results are presented as the mean ± SD of data from at least three independent experiments. Statistical analyses were performed using GraphPad Prism 6.0 (GraphPad Software Inc., CA, USA). Statistical significance was assessed by performing individual t-tests. P values lower than 0.05 were considered significant.

## 5. Conclusions

In this work, we demonstrated that tyrosine kinase receptor TIE-1 plays a critical role in cell proliferation and growth by modulating the PI3K/Akt signaling pathway in high-PI3K-expressing ovarian-cancer cells. Although further studies are still needed, our findings shed light on a possible mechanism to improve the poor prognosis of patients with high-PI3K-expressing ovarian cancer.

## Figures and Tables

**Figure 1 cancers-12-01705-f001:**
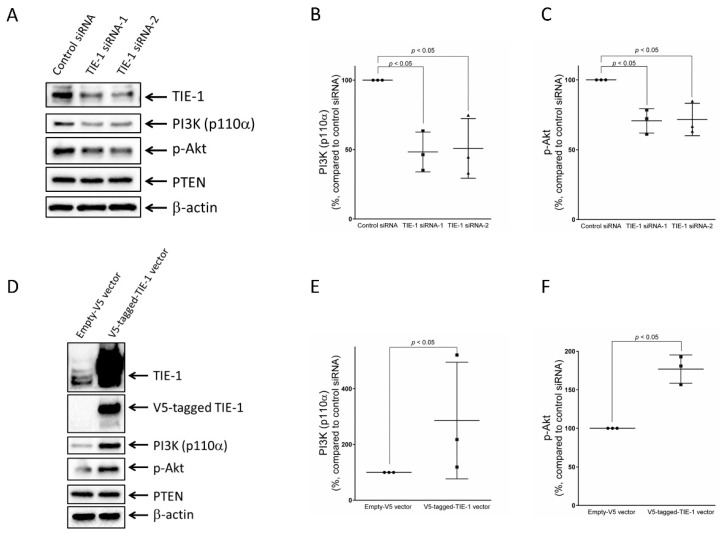
TIE-1 might signal through the PI3K/Akt pathway. (**A**) SKOV3 ovarian-cancer cells were transfected with 5 nM TIE-1 siRNA-1, TIE-1 siRNA-2, or control siRNA, for 72 h. Extracted cellular proteins were subjected to immunoblot analysis with antibodies against TIE-1, PI3K p110α, phospho-Akt, PTEN, and β-actin. Equal amounts of proteins were loaded in each lane. Three independent experiments were performed, and representative images are shown. Intensity of (**B**) PI3K p110α and (**C**) phospho-Akt protein expression quantified using Image Lab and shown as mean ± SD of three independent experiments; *, *p* < 0.05 compared with control siRNA group. (**D**) SKOV3 ovarian-cancer cells transfected with empty vector or V5-tagged TIE-1 vector for 72 h. Extracted cellular proteins subjected to immunoblot analysis with antibodies against TIE-1, V5, PI3K p110α, phospho-Akt, PTEN, and β-actin. Equal amounts of proteins were loaded in each lane. Three independent experiments were performed, and representative images are shown. Intensity of (**E**) PI3K p110α and (**F**) phospho-Akt protein expression quantified using Image Lab and shown as mean ± SD of three independent experiments; *, *p* < 0.05 compared with empty-vector group.

**Figure 2 cancers-12-01705-f002:**
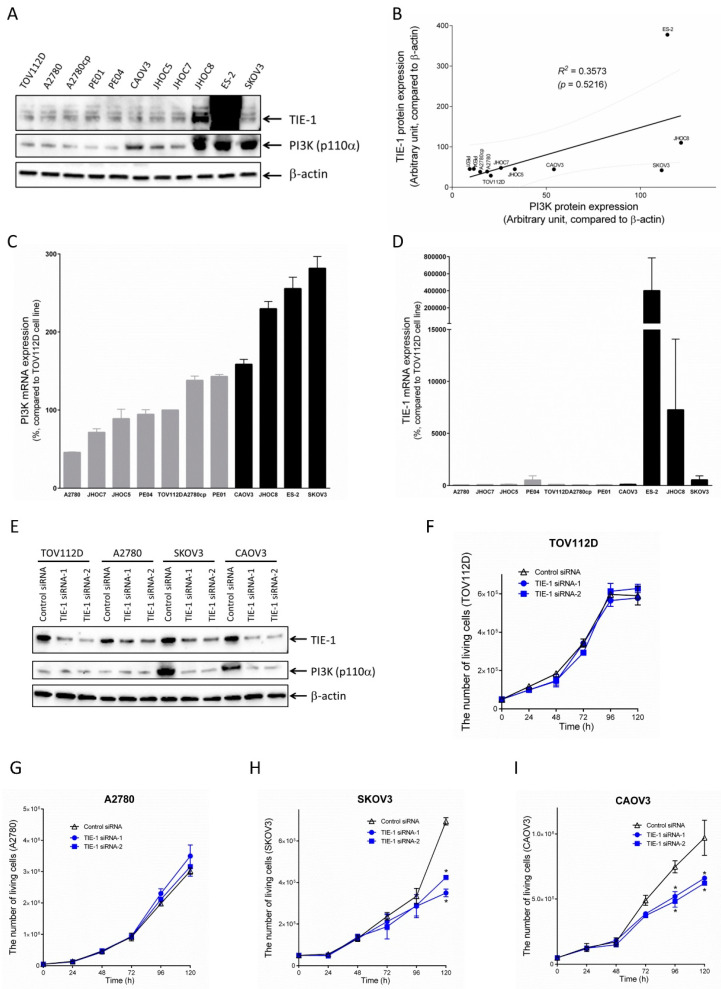
Inhibition of TIE-1 decreases cell growth in high-PI3K-expressing cell lines. (**A**) Extracted cellular proteins from indicated ovarian-cancer cell lines were subjected to immunoblot analysis with antibodies against TIE-1, PI3K p110α, and β-actin. Equal amounts of protein were loaded in each lane. Three independent experiments were performed, and representative images are shown. Intensity of TIE-1 and PI3K p110α protein expression quantified using Image Lab. (**B**) TIE-1 protein expression plotted with PI3K p110α protein expression, and *R*^2^ values determined by GraphPad prism. mRNA expression levels of (**C**) PI3K p110α and (**D**) TIE-1 in eleven ovarian-cancer cell lines confirmed by quantitative real-time PCR. (**E**) TOV112D, A2780, SKOV3, and CAOV3 ovarian-cancer cells transfected with 5 nM TIE-1 siRNA-1, TIE-1 siRNA-2, or control siRNA for 72 h. Extracted cellular proteins subjected to immunoblot analysis with antibodies against TIE-1, PI3K p110α, and β-actin. Equal amounts of proteins loaded in each lane. Three independent experiments were performed, and representative images are shown. (**F**) TOV112D, (**G**) A2780, (**H**) SKOV3, and (**I**) CAOV3 ovarian-cancer cells transfected with 5 nM TIE-1 siRNA-1, TIE-1 siRNA-2, or control siRNA, for up to 120 h. Number of living cells was counted. Values shown as mean ± SD (*n* = 3); *, *p* < 0.05 compared with control siRNA group.

**Figure 3 cancers-12-01705-f003:**
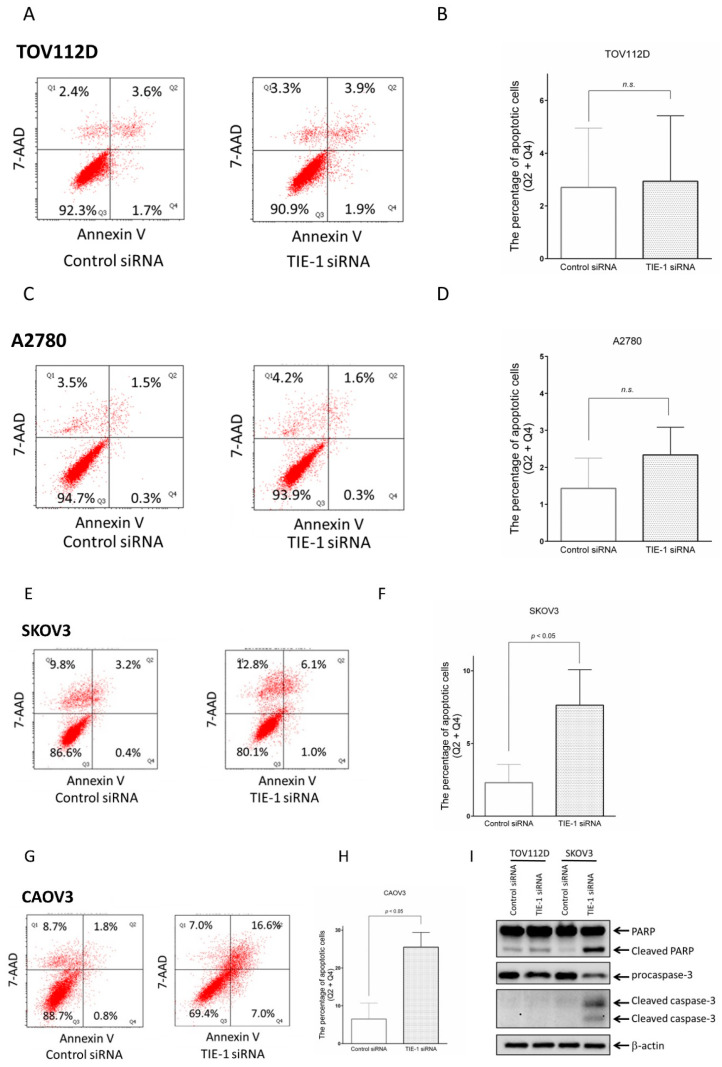
Inhibition of TIE-1 induces apoptosis in high-PI3K-expressing cell lines. (**A**) TOV112D cells were transfected with 5 nM TIE-1 siRNA or control siRNA for 72 h and assessed as described in Materials and Methods. Annexin V/7-annexin-apoptosis-detection (AAD) assays performed to analyze apoptotic cell death. (**B**) Percentage of apoptotic cells determined and presented as mean ± SD of three independent experiments. (**C**) A2780 cells transfected with 5 nM TIE-1 siRNA or control siRNA for 72 h and assessed as described in Materials and Methods. Annexin V/7-AAD assays performed to measure apoptotic cell death. (**D**) Percentage of apoptotic cells determined and presented as mean ± SD of three independent experiments. (**E**) SKOV3 cells transfected with 5 nM TIE-1 siRNA or control siRNA for 72 h and assessed as described in Materials and Methods. Annexin V/7-AAD assays performed to measure apoptotic cell death. (**F**) Percentage of apoptotic cells determined and presented as mean ± SD of three independent experiments; *, *p* < 0.05 compared with control siRNA group. (**G**) CAOV3 cells transfected with 5 nM TIE-1 siRNA or control siRNA for 72 h and assessed as described in Materials and Methods. Annexin V/7-AAD assays performed to analyze apoptotic cell death. (**H**) Percentage of apoptotic cells determined and presented as mean ± SD of three independent experiments; *, *p* < 0.05 compared with control siRNA group. (**I**) TOV112D and SKOV3 cells transfected with 5 nM TIE-1 siRNA or control siRNA for 72 h, and extracted cellular proteins subjected to immunoblot analysis with antibodies for poly ADP-ribose polymerase (PARP), procaspase-3, cleaved caspase-3, and β-actin. Equal amounts of proteins loaded in each lane. Three independent experiments were performed, and representative images are shown.

**Figure 4 cancers-12-01705-f004:**
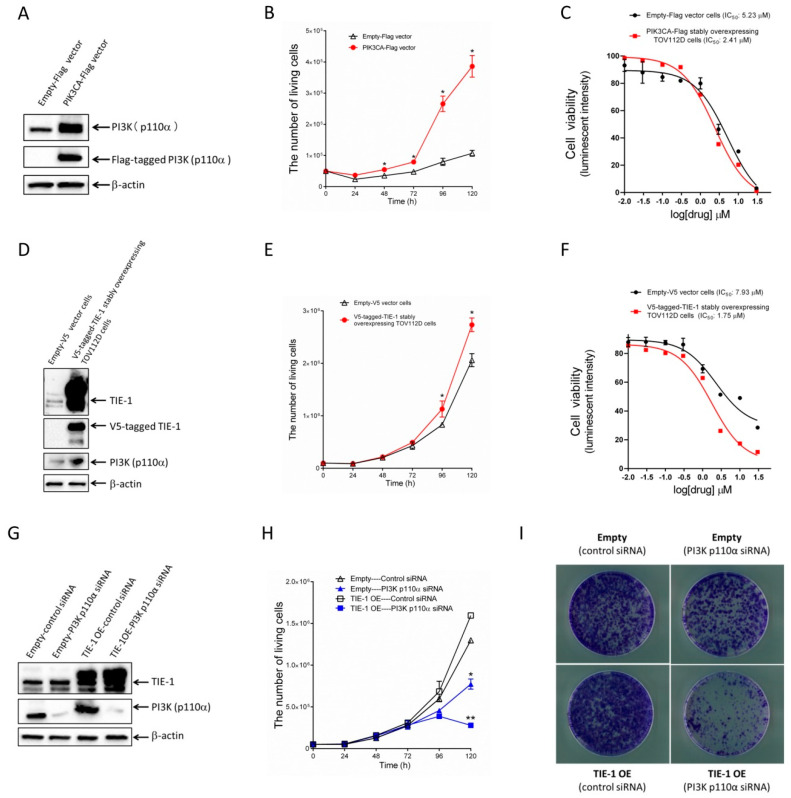
Increased TIE-1 expression induces TIE-1/PI3K-mediated cell growth in low-PI3K-expressing TOV112D ovarian-cancer cells. (**A**) TOV112D ovarian-cancer cells transfected with empty vector or PI3K–Flag vector for 72 h. Extracted cellular proteins subjected to immunoblot analysis with antibodies against PI3K p110α, Flag, and β-actin. Equal amounts of proteins loaded in each lane. Three independent experiments were performed, and representative images are shown. (**B**) TOV112D ovarian-cancer cells transfected with empty vector or PI3K–Flag vector for up to 120 h. Living cells were counted. Values shown as mean ± SD (*n* = 3); *, *p* < 0.05 compared with empty-vector group. (**C**) Empty Flag vector cells and TOV112D cells stably overexpressing Flag-tagged PI3K were treated with 0.01, 0.03, 0.1, 0.3, 1, 3, 10, or 30 µM of PI3K inhibitor GDC-0941 (Pictilisib). Cell viability determined using CellTiter-Glo luminescent cell-viability assay according to manufacturer’s protocol. Data shown (mean ± SD, *n* = 3) are percentages relative to cells treated with negative control. (**D**) Extracted cellular proteins from empty V5 vector cells and V5-tagged TIE-1 stably overexpressing TOV112D cells subjected to immunoblot analysis with antibodies against TIE-1, V5, PI3K p110α, and β-actin. Equal amounts of protein loaded in each lane. Three independent experiments were performed, and representative images shown. (**E**) V5-tagged TIE-1 stably overexpressing TOV112D cells or empty-vector cells seeded in six-well plates and incubated for up to 120 h. Living cells were counted. Values shown as mean ± SD (*n* = 3); *, *p* < 0.05 compared with empty-vector group. (**F**) Empty V5 vector cells and V5-tagged TIE-1 stably overexpressing TOV112D cells treated with 0.01, 0.03, 0.1, 0.3, 1, 3, 10, or 30 µM of PI3K inhibitor GDC-0941 (Pictilisib). Cell viability determined using CellTiter-Glo luminescent cell-viability assay according to manufacturer’s protocol. Data shown (mean ± SD, *n* = 3) are percentages relative to cells treated with negative control. (**G**) V5-tagged TIE-1 stably overexpressing TOV112D cells or empty V5 vector cells transfected with 5 nM PI3K p110α siRNA or control siRNA for 72 h. Extracted cellular proteins subjected to immunoblot analysis with antibodies against TIE-1, PI3K p110α, and β-actin. Equal amounts of proteins loaded in each lane. Three independent experiments were performed, and representative images are shown. (**H**) V5-tagged TIE-1 stably overexpressing TOV112D cells or empty vector cells transfected with 5 nM PI3K p110α siRNA or control siRNA for up to 120 h. Living cells were counted. Values shown as mean ± SD (*n* = 3); *, *p* < 0.05 compared with empty control siRNA group; **, *p* < 0.05 compared with TIE-1 control siRNA group. (**I**) V5-tagged TIE-1 stably overexpressing TOV112D cells or empty-vector cells transfected with 5 nM PI3K p110α siRNA or control siRNA for 9 days. Three independent experiments were performed. Imaging performed using ApeosPort-VI C4471 (Fuji Xerox, Japan), and representative images are shown.

**Figure 5 cancers-12-01705-f005:**
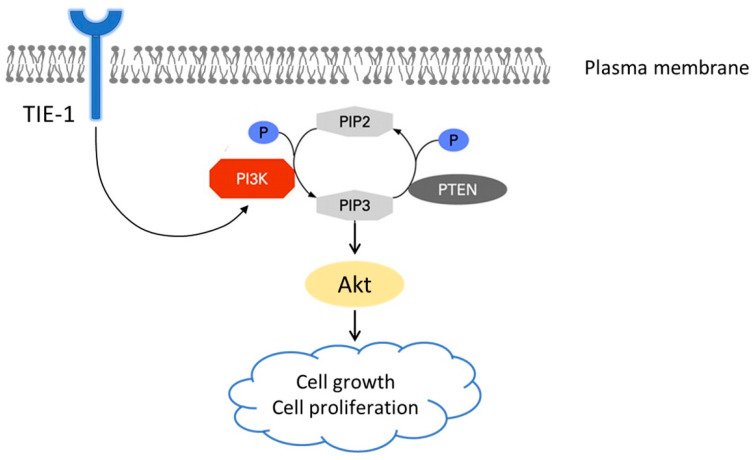
TIE-1 regulates PI3K expression and promotes PI3K-mediated cell proliferative phenotype. Proposed action mechanisms of TIE-1: high TIE-1 expression induces PI3K upregulation and PI3K-mediated cell proliferative phenotype.

**Table 1 cancers-12-01705-t001:** Histological type of 11 ovarian cancer cell lines.

Cell Name	Histological Type
TOV112D	Endometrioid
A2780	Non-serous
A2780cp	Non-serous
PE01	High grade serous carcinoma
PE04	High grade serous carcinoma
CAOV3	High grade serous carcinoma
JHOC5	Clear cell carcinoma
JHOC7	Clear cell carcinoma
JHOC8	Clear cell carcinoma
ES-2	Clear cell carcinoma
SKOV3	Non-serous

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
