# Peer review of "Potential of Tyrosine Kinase Receptor TIE-1 as Novel Therapeutic Target in High-PI3K-Expressing Ovarian Cancer"

_cancers, 2020, doi:10.3390/cancers12061705_

Round 1

Reviewer 1 Report

I reviewed the manuscript titled “Potential of tyrosine kinase receptor TIE-1 as a novel therapeutic target in PI3K high expressing ovarian cancer” by Zhang and colleagues. The manuscript describes the relation between Tie2 and PI3Kp110α in ovarian cancer cell lines. The authors identified PI3Kp110α low and high expressing cell lines and determined the effect of overexpression and knockdown of Tie2 on PI3Kp110α expression and cell proliferation and apoptosis. They also confirmed that the effect of Tie2 on proliferation and survival is mediated through PI3Kp110α by using a pharmacologic inhibitor of PI3Kp110α as well as silencing PI3Kp110α by siRNA.

I have the following concerns:

  1. The authors only used in vitro assays proliferation and colony survival. The authors should demonstrate the link between Tie2 and PI3Kp110α should in vivo in tumor xenografts of human cells in immunodeficient mice and in patients’ tumors.
  2. Figure 1A and D, the authors should show the level of total AKT.
  3. Figure 2B, show the p value.
  4. Figure 4: panels C and D and panels G and H should be combined each in one panel to show the changes in IC50 between control empty vector and overexpression ± inhibitor treatment.
  5. PI3Kp110α siRNA for 72h or 9 days are not convincing. The authors should use shRNA or CRISPR lentiviral vectors to stably knockdown/knockout PI3Kp110α.
  6. The manuscript needs extensive English language editing.

Reviewer 2 Report

Potential of tyrosine kinase receptor TIE-1 as a novel therapeutic target in PI3K high expressing ovarian cancer

by Xuewei Zhang et al

The manuscript reports experiments suggesting TIE-1’s role in regulation of PI3K-pathway in ovarian cancer cell lines. The authors demonstrate the effects of TIE-1 knockdown and overexpression in the PI3K, cell proliferation and apoptosis. They also show that PI3K-inhibitor modulates the effect of TIE-1 siRNA on the abovementioned cellular functions, which suggest that TIE-1 effect is PI3K-mediated.

The role of TIE-1 in cancer vascularization is well known. However, in ovarian cancer, only few functional studies have shown TIE-1’s role in growth and chemoresistance. As described by the authors, TIE-1 has been connected to PI3K-pathway in 3T3 mouse embryo cell model, while the connection between TIE-1 and PI3K in solid cancers has not been described before or this reviewer could not find any.

The text is well written and the experimental setting and data are solid. Still, there are some minor issues to be addressed.

  1. Line 16: chemorefracory ovarian cancer. Does all the used cell lines represent chemorefractory cancer? Please specify / demonstrate in the manuscript if claimed in the abstract.
  2. line 65-68: same result written twice is confusing
  3. and 68-71: same result written twice is confusing
  4. Figure 1: Why the other PI3K-pathway proteins were not studied? I would appreciate adding at least phospho-PI3K in the western blots for complete information of pw-activation.
  5. The ovarian cancer cell lines used are of varying histotypes but mainly of type-1 ovarian cancer (clear cell, endometroid). Due to the fact that distinct histotypes have diverse origin and molecular pathogenic mechanisms, it would be useful to let a reader know about the ovca subtypes that cells represent (Caov3 HGSOC, SKOV3 clear cell or endometroid, TOV112D and A2780 endometroid).
  6. Cell lines:
    • As two distinct ES-2/ES2 cell line exist, one ovarian CCC, one Ewing Sarcoma, please pay attention to use correct abbreviation throughout the manuscript and figures. Now both abbreviations are used.
    • SKOV-3 has PI3K amplification (Domcke et al DOI: 10.1038/ncomms3126). Please discuss the results based on that. Might be useful to remove this cell line in the correlation analysis (Fig 2B) due to the genetic alteration.
  7. Fig 2D, are error bars of JHOC8 missing due to the y-axis break? Unifying the order of cell lines in Fig 2C and D would be helpful for a reader.
  8. Results 2.2. line 96 and Fig 2 E-I. Neither of the two cell lines with very high TIE-1 and PI3K level (JHOC8 and ES-2) were used although they would be very interesting to evaluate. I understand ES-2 due to growth issue (Supplemental data) but is there a specific reason why JHOC8 was not used in further studies?
  9. Results 2.3 and Fig 3. Does siTIE-1 have similar effect on PARP / caspase in CAOV-3 similar to SKOV-3? If yes, the result would show reproducibility of the mechanism over the TIE-1 / PI3K high cell lines.
  10. Lines 164-171: This paragraph is difficult to read and is recommended to be simplified. It may not be necessary to repeat the comparison to empty vector many times. Please consider the issue through the whole results section.
  11. Conclusion: Please replace “carcinogenic agent” to other more accurate term for a native oncogenic protein.
  12. Supplemental Figure 3 has confusing labels (e.g. Fig 1A followed by 1D etc.). Please finalize.

Round 2

Reviewer 1 Report

The authors addressed my earlier concerns.